# A unicellular relative of animals generates a layer of polarized cells by actomyosin-dependent cellularization

Omaya Dudin[1†]*, Andrej Ondracka[1†], Xavier Grau-Bové[1,2], Arthur AB Haraldsen[3], Atsushi Toyoda[4], Hiroshi Suga[5], Jon Bråte[3], Iñaki Ruiz-Trillo[1,6,7]*

[1]Institut de Biologia Evolutiva (CSIC-Universitat Pompeu Fabra), Barcelona, Spain; [2]Department of Vector Biology, Liverpool School of Tropical Medicine, Liverpool, United Kingdom; [3]Section for Genetics and Evolutionary Biology (EVOGENE), Department of Biosciences, University of Oslo, Oslo, Norway; [4]Department of Genomics and Evolutionary Biology, National Institute of Genetics, Mishima, Japan; [5]Faculty of Life and Environmental Sciences, Prefectural University of Hiroshima, Hiroshima, Japan; [6]Departament de Genètica, Microbiologia i Estadística, Universitat de Barcelona, Barcelona, Spain; [7]ICREA, Barcelona, Spain

**Abstract** In animals, cellularization of a coenocyte is a specialized form of cytokinesis that results in the formation of a polarized epithelium during early embryonic development. It is characterized by coordinated assembly of an actomyosin network, which drives inward membrane invaginations. However, whether coordinated cellularization driven by membrane invagination exists outside animals is not known. To that end, we investigate cellularization in the ichthyosporean *Sphaeroforma arctica*, a close unicellular relative of animals. We show that the process of cellularization involves coordinated inward plasma membrane invaginations dependent on an actomyosin network and reveal the temporal order of its assembly. This leads to the formation of a polarized layer of cells resembling an epithelium. We show that this stage is associated with tightly regulated transcriptional activation of genes involved in cell adhesion. Hereby we demonstrate the presence of a self-organized, clonally-generated, polarized layer of cells in a unicellular relative of animals.

**\*For correspondence:**
omaya.dudin@outlook.com (OD);
inaki.ruiz@ibe.upf-csic.es (IR-T)

[†]These authors contributed equally to this work

**Competing interests:** The authors declare that no competing interests exist.

## Introduction

Cellularization of a coenocyte — a multinucleate cell that forms through sequential nuclear divisions without accompanying cytokinesis — is a specialized form of coordinated cytokinesis that results in cleavage into individual cells. Cellularization commonly occurs during development of animals, plants and unicellular eukaryotes (*Hehenberger et al., 2012*; *Mazumdar and Mazumdar, 2002*). Despite the similarities, distinct mechanisms are involved in the cellularization of these coenocytes. During endosperm development in most flowering plants, coenocytes cellularize through cell wall formation around individual nuclei, forming a peripheral layer of cells surrounding a central vacuole (*Hehenberger et al., 2012*). This is coordinated by the radial microtubule system and is dependent on several microtubule-associated proteins (*Pignocchi et al., 2009*; *Sørensen et al., 2002*). In apicomplexan parasites, cytokinesis of multinucleate schizonts occurs by budding, which is driven by a polarized microtubule scaffold (*Francia and Striepen, 2014*).

In contrast, in a model animal coenocyte, the syncytial blastoderm of the fruit fly *Drosophila melanogaster*, cellularization is accomplished through plasma membrane invaginations around equally spaced, cortically positioned nuclei (*Farrell and O'Farrell, 2014*; *Mazumdar and Mazumdar, 2002*).

This process relies on extensive membrane remodeling (*Lecuit, 2004*; *Sokac and Wieschaus, 2008*; *Figard et al., 2016*) regulated by zygotically transcribed genes (*Schweisguth et al., 1990*; *Schejter and Wieschaus, 1993*; *Postner and Wieschaus, 1994*; *Hunter and Wieschaus, 2000*; *Lecuit et al., 2002*) and driven by microtubules and a contractile actomyosin network (*Mazumdar and Mazumdar, 2002*). It depends on several actin nucleators, such as the Arp2/3 complex (*Stevenson et al., 2002*) and formins (*Afshar et al., 2000*). It also requires multiple actin-binding proteins, including myosin II (*Royou et al., 2002*), which mediates actin cross-linking and contractility, as well as septins (*Adam et al., 2000*; *Cooper and Kiehart, 1996*), cofilin (*Gunsalus et al., 1995*) and profilin (*Giansanti et al., 1998*). In addition, it depends on cell-cell adhesion proteins including cadherin, and alpha- and beta-catenin (*Hunter and Wieschaus, 2000*; *Wang et al., 2004*). This coordinated cellularization results in the formation of a single layer of polarized epithelial tissue, also known as cellular blastoderm (*Mazumdar and Mazumdar, 2002*). This actomyosin-dependent cellularization is common in early insect embryos and also commonly observed in germline development of many animals (*Haglund et al., 2011*), such as the nematode *C. elegans* (*Priti et al., 2018*), however, whether this mechanism of cellularization is found outside animals, remains unknown.

Among holozoans — a clade that includes animals and their closest unicellular relatives (*Figure 1A*) — ichthyosporeans are the only known lineage besides animals that forms coenocytes during their life cycles (*Mendoza et al., 2002*; *de Mendoza et al., 2015*). All characterized ichthyosporeans proliferate through rounds of nuclear divisions within a cell-walled coenocyte, followed by release of newborn cells (*Marshall and Berbee, 2011*; *Marshall and Berbee, 2013*; *Suga and Ruiz-Trillo, 2013*; *Whisler, 1968*). We have previously suggested that they undergo cellularization (*Ondracka et al., 2018*; *Suga and Ruiz-Trillo, 2013*). However, at present nothing is known about the ichthyosporean cellularization, and whether it involves animal-like mechanisms.

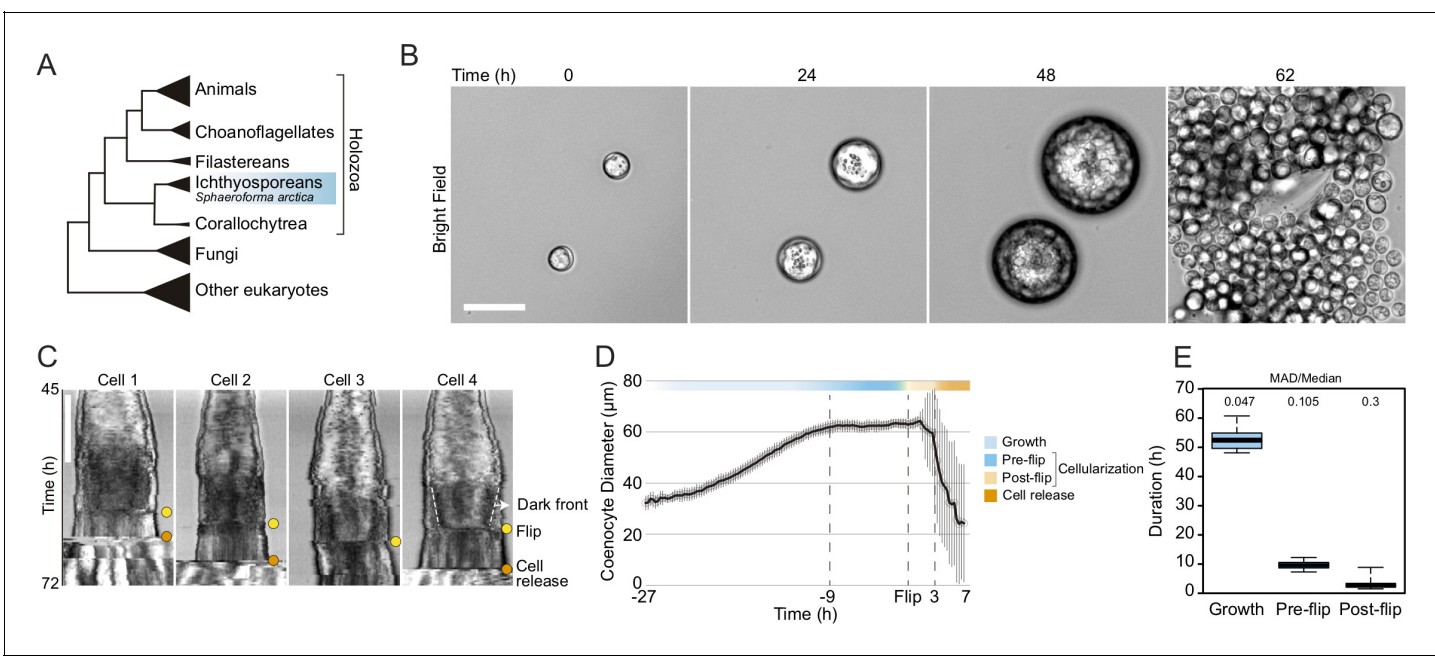

**Figure 1.** Cellularization dynamics in *Sphaeroforma arctica*. (A) Phylogenetic position of the ichthyosporean *Sphaeroforma arctica* in the tree of life. (B) Time-lapse images of the life-cycle of *S. arctica* show cell-size increase prior to cellularization and release of new-born cells. Associated with *Video 2*. Bar, 50 μm. (C) Kymographs of 4 distinct cells undergoing cellularization with the time of flip (yellow) and cell release (orange) indicated for each. An example of the advancing dark front is shown for cell 4. Bar, 50 μm. (D) Mean cell diameter over time of 65 cell traces aligned to Flip reveals distinct cell stages: Growth, Pre and post-flip and cell release. (E) Duration of growth, cellularization and post-flip represented as box-plots (N˚cells > 100). MAD (Median absolute deviation) over median is used as a measure of variability.

The online version of this article includes the following figure supplement(s) for figure 1:

**Figure supplement 1.** Cellularization dynamics in *Sphaeroforma arctica*.

Here, we comprehensively characterized cellularization in the ichthyosporean *Sphaeroforma arctica,* in which we have previously shown that coenocytic cycles can be synchronized (*Ondracka et al., 2018*). We used imaging, transcriptional profiling, and pharmacological inhibition, to study the gene expression dynamics, morphological rearrangements, and mechanisms of cellularization. We found that cellularization is accomplished by inward plasma membrane invaginations driven by an actomyosin network, forming a polarized layer of cells. Time-resolved transcriptomics revealed sharply regulated expression of cell polarity and cell adhesion genes during this stage. Finally, we show that this process depends on actin nucleators and Myosin II, and we reveal the temporal order of the actomyosin network assembly. Together, these findings establish that cellularization of ichthyosporeans proceeds by mechanism conserved between animals and ichthyosporeans.

## Results

### Growth and cellularization in *S. arctica* are temporally separated stages of the coenocytic cycle

To determine the timing of cellularization in synchronized cultures, we established long-term live imaging conditions. Individual coenocytes exhibit growth in cell size until approximately 60 hr, after which they undergo release of newborn cells (*Figure 1B*, *Video 1*). This was consistent with previous results in bulk cultures (*Ondracka et al., 2018*), confirming that our experimental setup for long-term live imaging faithfully reproduces culture growth. However, by measuring the diameter of the coenocytes, we observed that newborn cell release occurred with somewhat variable timing (*Figure 1—figure supplement 1A*, *Video 1*).

Time-lapse imaging revealed that prior to cell release, the coenocytes darken along the periphery, and the dark front begins to advance towards the center (*Video 2*). Afterwards, we observed an abrupt internal morphological change in the coenocyte, when the front disappears. We termed this event 'flip' (*Video 2*). The flip occurred in all the coenocytes and can be reliably detected on kymographs (*Figure 1C*). Aligning individual coenocyte size traces to this specific common temporal marker, we observe that coenocytes stop growing approximately 9 hr before the flip (*Figure 1D and E* and *Figure 1—figure supplement 1B*). This shows that the growth stage and cellularization are temporally separated. The cellularization can be divided into a temporally less variable pre-flip phase (~9 hr) and a variable post-flip phase (*Figure 1E* and *Figure 1—figure supplement 1B and C*). Taken together, these results show that growth and cellularization form temporally distinct stages of the coenocytic cycle of *S. arctica.* This provides a temporal framework in which to characterize phenotypically distinct stages of cellularization.

### The cortical actin network establishes sites of membrane invagination and generates a polarized layer of cells during cellularization

To assess whether cellularization in *S. arctica* involves encapsulation of nuclei by plasma membranes, we imaged the plasma membrane using live time-lapse imaging in the presence of the lipid dye FM4-64 (*Betz et al., 1996*). We observed a rapid increase in FM4-64 intensity at the periphery of the coenocyte 30 min prior to flip (*Figure 2A*, panel II, *Videos 3* and *4*). The plasma membrane invagination sites formed at the periphery and progressed synchronously from the outside toward the center of the coenocyte, forming polarized, polyhedral cells (*Figure 2A*, panels II-V, *Videos 3* and *4*). Similar

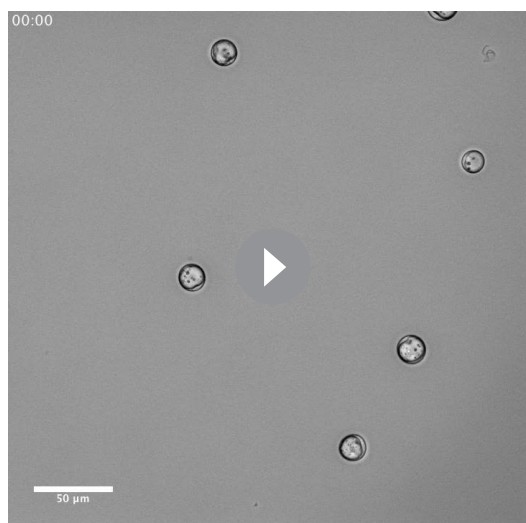

**Video 1.** Time lapse of synchronized cells of *S. arctica* obtained with epifluorescent microscopy. Time interval between frames is 20 min. The movie is played at 7fps. Four distinct cells can be seen undergoing a full life-cycle with the release of new born cells. Bar, 50 μm.
https://elifesciences.org/articles/49801#video1

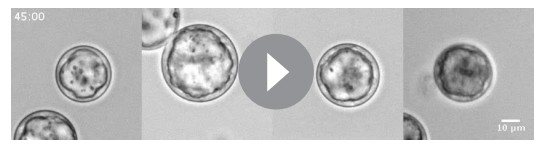

**Video 2.** Time lapse of synchronized cells of *S. arctica* obtained with epifluorescent microscopy. Time interval between frames is 15 min. The movie is played at 7fps. Three cells can be seen undergoing flip prior to cell release whereas one cell undergo 'flip' without cell release during the course of the experiment. The exact timing of cell release is pointed out for each cell. Bar, 50 µm.

https://elifesciences.org/articles/49801#video2

to cellularization in *Drosophila* coenocytes (*Lecuit and Wieschaus, 2000*; *Lecuit et al., 2002*), we observe different rates of plasma membrane invaginations throughout the process. During the first 10–15 min, the invaginations advance at a rate of about 0,3 µm/min (*Figure 2—figure supplement 1A*). This period is followed by a rapid phase where membrane invaginates at a rate of about 0,9 µm/min leading to flip (*Figure 2—figure supplement 1A*). Finally, following flip, cells lost their polyhedral shape and became round, suggesting that they were no longer attached to each other (*Figure 2A*, panel VI, *Videos 3* and *4*).

In animal coenocytes, plasma membrane invagination is associated with dynamic organization of the actomyosin cytoskeleton (*Mazumdar and Mazumdar, 2002*). To investigate actin dynamics during cellularization, we took advantage of the timeline described above and imaged coenocytes that were fixed and stained for actin and nuclei (using phalloidin and DAPI, respectively) at different time points during cellularization (*Figure 2B*, *Figure 2—figure supplement 1B and C*). During growth, actin localized exclusively as small patches at the surface of the coenocyte (*Figure 2—figure supplement 1B*, panel I). Only at the onset of cellularization, multiple actin patches increased in size to form actin nodes (*Figure 2B*, panel I and II, and *Figure 2—figure supplement 1B*, panel II). This phase was followed by cortical compartmentalization surrounding the nuclei through gradual formation of an actin filament network solely at the cortex of the coenocyte (*Figure 2B*, panel III, and *Figure 2—figure supplement 1B*, panels III and IV). Following this cortical compartmentalization, a layer of cells was transiently formed by inward growth of the actin filaments from the cortex (*Figure 2B*, panels IV and *Figure 2—figure supplement 1B*, panels V and VI). During this stage, the actin signal intensity was uneven and higher on the internal side (*Video 5*), and nuclei were localized close to the cortex, indicating that cells are polarized (*Figure 2B*, panel IV, *Video 5*). These polarized cells progressively grew towards the center of the coenocyte to fill the cavity (*Figure 2—figure supplement 1B*, panel VII). After flip, similar to the plasma membrane organization mentioned above, the layer of cells was reorganized to form spherical cells (*Figure 2B*, panel VI, and *Figure 2—figure supplement 1B*, panel VIII).

To determine the order of actin organization and plasma membrane invaginations, we assessed the localization of both actin and plasma membrane in fixed samples using phalloidin and a fixable analog of FM4-64 (FM4-64FX). We found that the cortical actin network formed prior to the appearance of the membrane dye (*Figure 2C*, panel II).

The intensity of FM4-64FX labeling also increased and co-localized with the underlying actin network around the time of plasma membrane invagination (*Figure 2C*, panels III and IV). This suggests that the cortical actin network determines the site of plasma membrane invagination.

Finally, to determine the timing of cell wall formation, we stained the cells with calcofluor. We observed that labeling co-localized with the membrane marker FM4-64FX around individual cells in the post-flip stage (*Figure 2—figure supplement 1D*). This shows that the newborn cells already build the cell wall before the release, as was suggested previously in other *Sphaeroforma* species (*Marshall and Berbee, 2013*).

In early insect embryos, cellularization of the syncytial blastoderm occurs through actin-dependent invagination of the plasma membrane. Here, we demonstrate that the cellularization of the ichthyosporean coenocyte also involves active actin reorganization and membrane invagination (slow and fast phases of membrane invagination rates) at the site of actin cytoskeleton. Additionally, this results in the transient formation of a polarized layer of cells with an internal cavity that morphologically resembles simple epithelial structures.

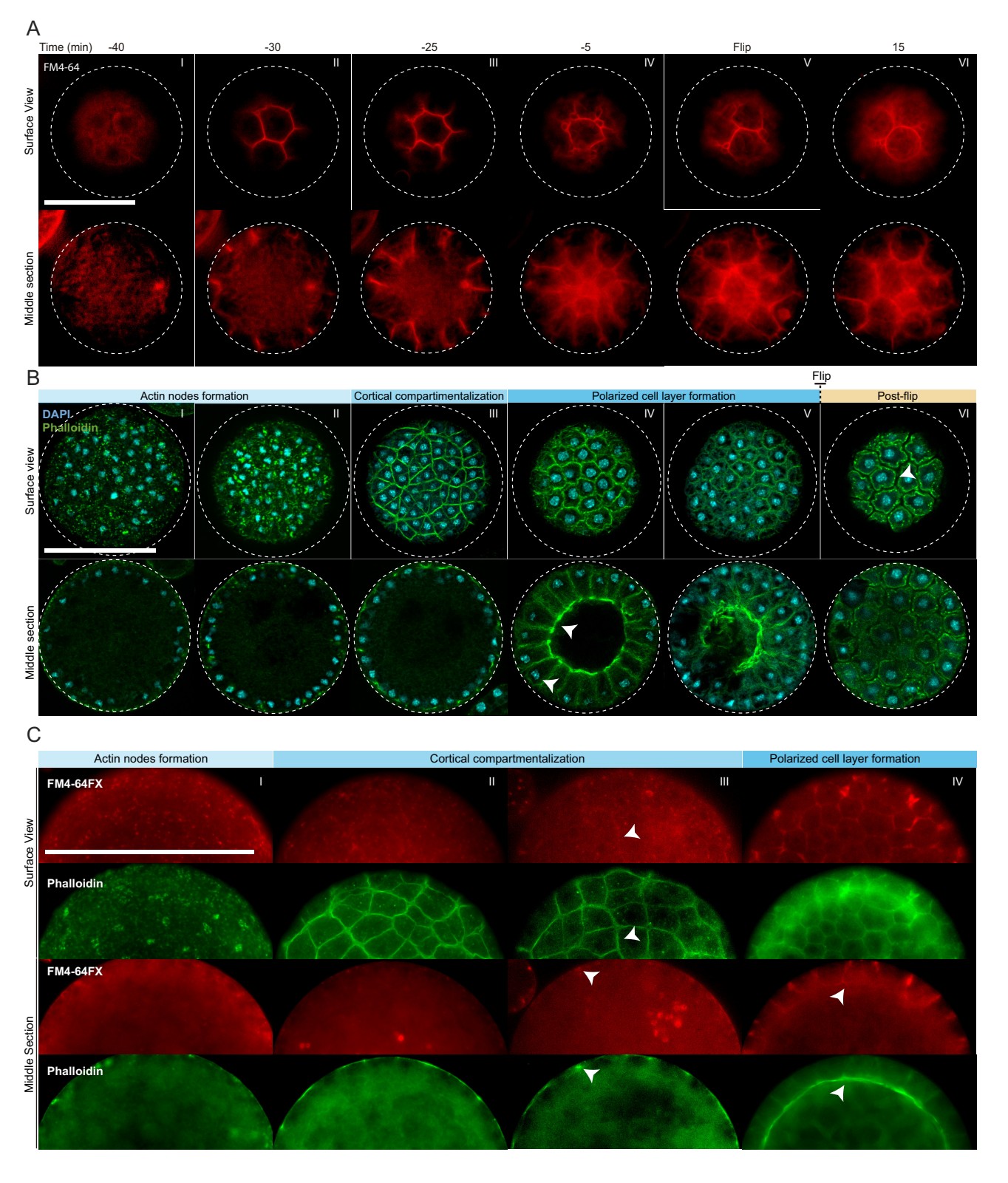

**Figure 2.** Actin cytoskeleton and plasma membrane dynamics during cellularization of *S. arctica*. (**A**) Dynamics of plasma membrane invaginations during cellularization. Live-cells, pre-grown for 58 hr, were stained with FM4-64 (10 μM) and imaged using epifluorescent microscopy with a 5 min interval. Bar, 50 μm. (**B**) Spatio-temporal organization of the actin cytoskeleton, nuclei and cells during cellularization. Synchronized cells of *S. arctica*, pre-grown for 48 hr, were fixed every hour for 14 hr and stained with phalloidin and DAPI to reveal cytoskeletal dynamics during cellularization. All cells

*Figure 2 continued on next page*

*Figure 2 continued*

were imaged using confocal microscopy. In panel IV, arrows indicate higher actin signal intensity on the internal side and that nuclei are localized close to the cortex indicating that the layer of cells is polarized. Bar, 50 µm. (C) Actin network is established prior to plasma membrane invaginations. Synchronized cells of *S. arctica*, pre-grown for 54 hr, were fixed every 2 hr for 10 hr and stained with both the membrane dye FM4-64FX and phalloidin. Arrows show sites of colocalization between both markers at the onset of plasma membrane invaginations. Bar, 50 µm.

The online version of this article includes the following figure supplement(s) for figure 2:

**Figure supplement 1.** Actin cytoskeleton, plasma membrane and cell-wall dynamics during cellularization of *S. arctica*.

## Cellularization is associated with extensive sequential transcriptional waves and is associated with evolutionarily younger transcripts

To gain insight into the regulation of the cellularization of *S. arctica*, we sequenced the expressed mRNAs of synchronized cultures with high temporal resolution, and comprehensively analyzed the dynamics of transcription, alternative splicing, and long intergenic non-coding RNAs (lincRNAs). Because the published genome assembly of *S. arctica* (*Grau-Bové et al., 2017*) was fragmented and likely resulted in incomplete gene models, we first re-sequenced the genome combining the Illumina technology with the PacBio long read sequencing technology. The final assembly sequences comprised 142,721,209 bp, and the metrics were greatly improved compared to the previous assembly (*Grau-Bové et al., 2017*) (*Figure 3—source data 1*). *Ab initio* gene annotation resulted in the discovery of novel ORFs due to the absence of repetitive regions in the previous assembly. RNA-seq data was used to improve the ORF prediction, to define the 5' and 3' untranslated regions, and to discover lincRNAs. In total, 33,682 protein coding genes and 1071 lincRNAs were predicted using this combined pipeline (see Materilas and methods). This final transcriptome assembly was used as the reference transcriptome for further analysis.

To perform the time-resolved transcriptomics, we isolated and sequenced mRNA from two independent synchronized cultures at 6 hr time intervals during the entire coenocytic cycle, encompassing time points from early 4-nuclei stage throughout growth and cellularization stages until the release of newborn cells (*Figure 3—figure supplement 1A*).

We first analyzed transcript abundance during the time series. The majority of the transcriptome (20,196 out of 34,753 predicted protein-coding and lincRNA genes) was transcribed at very low levels (mean expression throughout the time course <0.5 tpm [transcripts per million]) and were removed for the subsequent clustering analysis. Clustering of transcript abundance data from both biological replicates revealed a clear separation between the transcriptomes of the growth stage (12 hr to 42 hr time points) and the cellularization stage (48 hr to 66 hr time points) (*Figure 3A*). Furthermore, we observed that the transcriptome patterns between replicates 1 and 2 were shifted by 6 hr from 48 hr onwards (*Figure 3A*), presumably due to differences in temperature and conditions influencing the kinetics of the coenocytic cycle. We thus adjusted the time of the second replicate by 6 hr according to the clustering results, although we emphasize that the clustering analysis did not depend on time. Among the expressed transcripts (defined as mean expression levels higher than 0.5 tpm across all samples; in total 13,542 coding genes and 1015 lincRNAs), consensus clustering using Clust (*Abu-Jamous and Kelly, 2018*) extracted 9 clusters of co-expressed transcripts with a total of 4441 protein coding genes (*Figure 3B*), while the rest of the transcripts were not assigned to any co-expression cluster. The assigned cluster membership was robust to using either of the replicates or averaging (*Figure 3—figure supplement 1B*). Visualization by heatmap and t-SNE plot separated the clusters into two meta-clusters containing the growth stage (clusters 1–3, totaling 2197 genes) and cellularization stage (clusters 4–9, totaling 2314 genes) clusters (*Figure 3C and D*). Among the cellularization clusters, we obtain three main

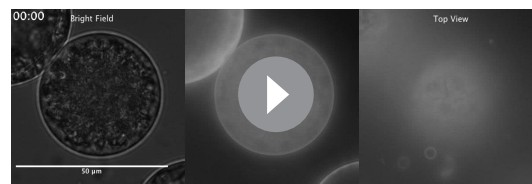

**Video 3.** Time lapse of synchronized cells of *S. arctica* stained with the plasma membrane dye FM4-64 and obtained with epifluorescent microscopy. Time interval between frames is 5 min. The movie is played at 7fps. Plasma membrane invaginations can be seen occurring from the outside inwards for approximately 40 min prior to flip. Bar, 50 µm.

https://elifesciences.org/articles/49801#video3

**Video 4.** Time lapse of plasma membrane dynamics during cellularization of *S. arctica*.
https://elifesciences.org/articles/49801#video4

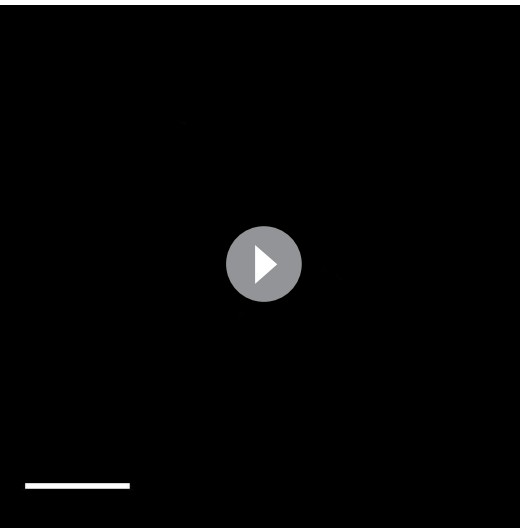

**Video 5.** Z-projection of the spatial organization of the polarized layer of cells during cellularization of *S. arctica* obtained with confocal microscopy. The cell is stained with phalloidin and DAPI. The movie is played at 7fps. A polarized layer of cells can be seen. Bar, 10 µm.
https://elifesciences.org/articles/49801#video5

clusters of genes sharply activated during cellularization (clusters 5, 6 and 7), which contain transcripts expressed at early, mid and late cellularization, respectively (*Figure 3B and D*). Altogether, this shows that cellularization is associated with extensive sharp transcriptional activation in multiple temporal waves, in total affecting 17% of the expressed transcriptome.

In parallel to transcript abundance dynamics, we also assessed alternative splicing (AS) across the time series. This analysis identified 2022 genes affected by intron retention (12.9% of all intron-bearing genes, totaling 4310 introns), 914 by exon skipping (12.3% of genes with >2 exons, 1206 exons) and 44 with mutually exclusive exons (0.7% of genes with >3 exons, involving 118 exon pairs) in all samples (*Figure 3—figure supplement 1C and D*). Overall, neither the number of AS events nor the number of genes affected vary dramatically along the *S. arctica* growth cycle (*Figure 3—figure supplement 1E-J*). Analysis of AS events over time did not yield any discernible global dynamics, although we found a small number of events differentially present between the growth and cellularization stages (*Figure 3—figure supplement 1K*).

Interestingly, skipped exons were more likely to be in-frame (38.63%, compared to 30.33% of in-frame exons in genes with >2 exons, p=4.34e-05, Fisher's exact test) and yield non-truncated transcripts, a phenomenon commonly observed in animal transcriptomes but not in transcriptomes of other unicellular eukaryotes (*Grau-Bové et al., 2018*). The transcripts affected by such in-frame exon skipping events are enriched in biological processes such as regulation of multicellular organismal processes, assembly of the focal adhesion complex and cell growth (*Figure 3—figure supplement 1L*). Overall, although pervasive, alternative splicing likely does not play a major role in regulation of the coenocytic cycle and cellularization of *S. arctica*.

Additionally, we analyzed the dynamics of lincRNA expression. Overall, lincRNAs represent ~3% of total transcript abundance across the time series (*Figure 4—figure supplement 1A*). Among the long non-coding RNAs, 70 lincRNA transcripts clustered with coding genes into temporally co-expressed clusters (*Figure 4A*). Sequence homology searches revealed that 24 of the *S. arctica* lncRNAs were conserved in distantly related ichthyosporean species such as *Creolimax fragrantissima*, *Pirum gemmata* and *Abeoforma whisleri* (estimated to have diverged ~500 million years ago; *Parfrey et al., 2011*). This is a remarkable depth of conservation, since animal lincRNAs are not conserved between animal phyla (*Bråte et al., 2015*; *Gaiti et al., 2015*; *Hezroni et al., 2015*). Other lincRNAs were either specific to *S. arctica* (511) or conserved only in closely related *Sphaeroforma* species (536). Comparison of lincRNAs by degree of conservation showed no notable difference in GC content (*Figure 4—figure supplement 1B*) or dynamics of expression during the coenocytic cycle (*Figure 4—figure supplement 1C*). However, we found that conserved lincRNAs were on average longer than non-conserved ones (*Figure 4—figure supplement 1D*) and, strikingly, expressed at much higher levels (*Figure 4B*). Among the 24 deeply conserved lincRNAs, five clustered in the temporally co-expressed clusters, which is higher than expected by chance (p=0.0166, Fisher's exact test). Among these, three belonged to the cellularization clusters, including, lincRNA asmbl_31839,

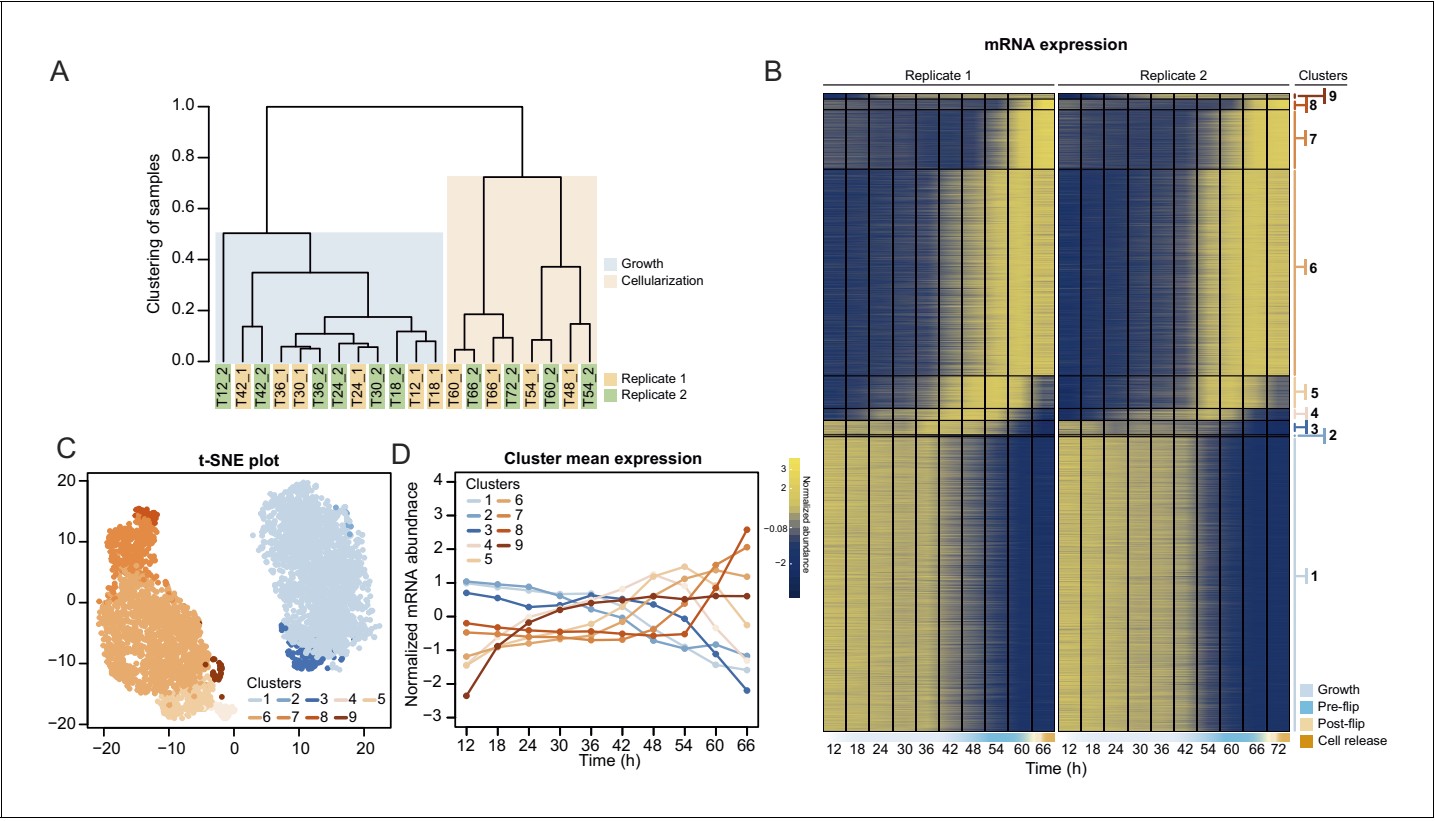

**Figure 3.** Transcriptional dynamics across the *S. arctica* life cycle. (**A**) Hierarchical clustering of time point samples by Euclidian distance of spearman correlation coefficient. The sample T48_2 is missing due to technical reason. (**B**) A heatmap of 4441 coding genes that were clustered into nine clusters. (**C**) A t-SNE plot of clustered genes. (**D**) Mean expression profile of each gene expression cluster.

The online version of this article includes the following source data and figure supplement(s) for figure 3:

**Source data 1.** Metrics of the *Sphaeroforma arctica* genome assemblies.
**Source data 2.** Transcript abundance (in tpm) of all the *S. arctica* transcripts.
**Source data 3.** Normalized transcript abundance of the expressed *S. arctica* transcripts (mean tpm > 0.5).
**Source data 4.** Table of transcripts per cluster membership.
**Figure supplement 1.** Nuclear content distribution, clustering and alternative splicing analysis.

which has a remarkably high sequence similarity with its homologs from other ichthyosporeans (*Figure 4—figure supplement 1E*). Furthermore, its transcriptional regulation is independent of the transcriptional regulation of its neighboring coding genes (located within 3 kb) (*Figure 4C*). In summary, we discovered deeply conserved lincRNAs that are expressed at high levels and are transcriptionally activated during cellularization.

Finally, to assess the evolutionary origins of the co-expression clusters, we used a phylostratigraphic analysis to classify genes into evolutionary gene age groups. We carried out orthology analysis of the predicted *S. arctica* proteome along with 30 representative species from the eukaryotic tree of life to identify 'orthogroups' (i.e. groups of putative orthologs between species). *S. arctica* protein-coding genes clustered in 6149 orthogroups representing 12,527 genes; the rest of the genes did not have an ortholog outside *S. arctica*.

Next, we inferred the age of each gene using Dollo parsimony (*Csurös, 2010*) to classify them into phylostrata (sets of genes with the same phylogenetic origin) (*Figure 4D*). Analysis of gene expression by phylostrata revealed a trend toward more variable expression throughout the coenocytic cycle in younger genes (*Figure 4E*), although their mean expression levels were lower (*Figure 4—figure supplement 2A*). Such a correlation has been observed in animal development, where developmentally regulated genes tend to be of younger origin (*Domazet-Lošo and Tautz, 2010*). Analysis of enrichment of gene phylostrata in each gene expression cluster (*Figure 4—figure supplement 2B*) showed that the growth clusters are enriched for pan-eukaryotic genes. In contrast, we

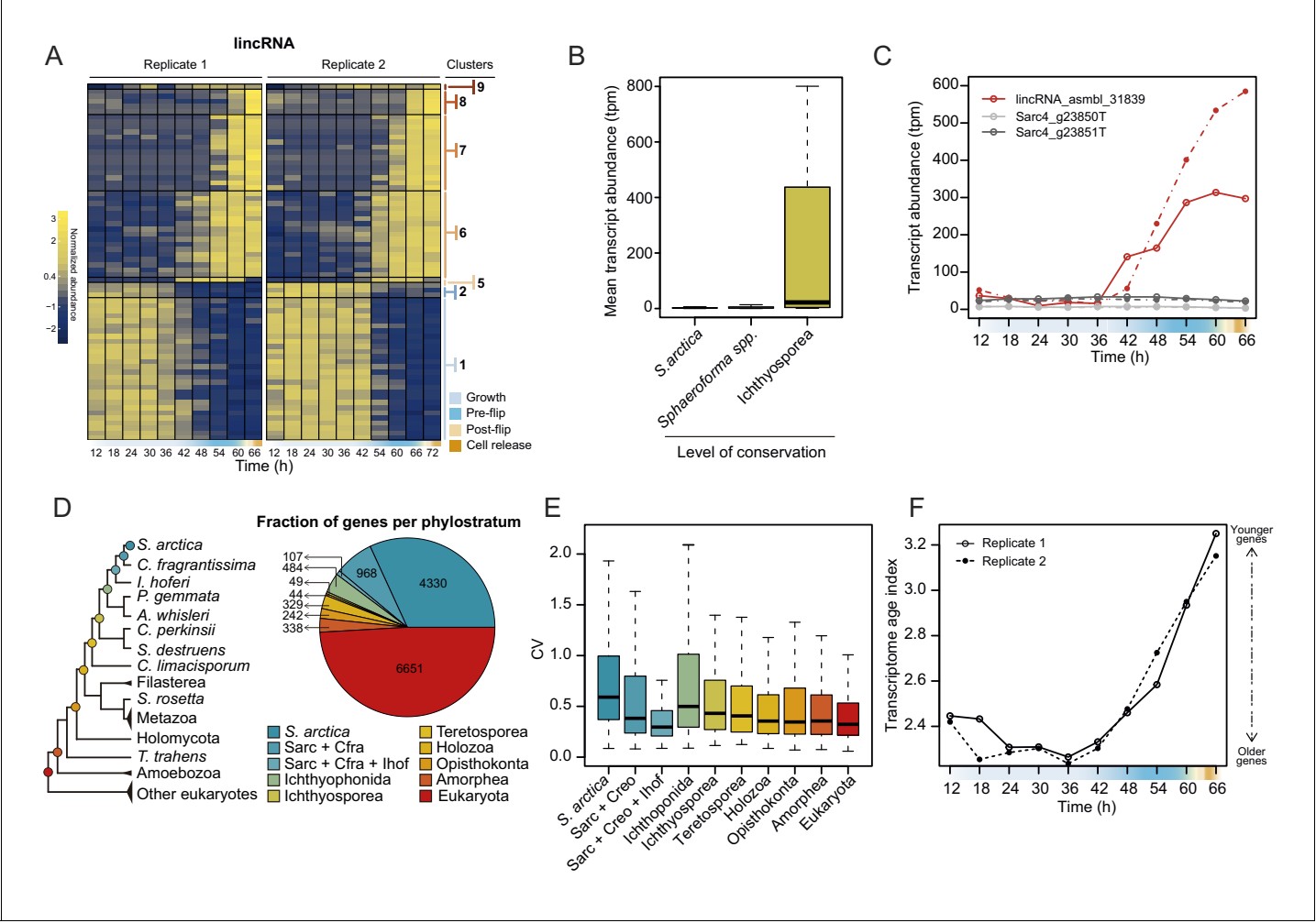

**Figure 4.** Dynamics of lincRNAs, alternative splicing, and gene phylostrata. (**A**) A heatmap of 70 long intergenic non-coding RNAs (lincRNAs) that co-cluster with coding genes. (**B**) mean expression level of the lincRNAs, binned by degree of conservation. (**C**) Expression of the conserved lincRNA, lincRNA: asmbl_31839 and the two coding genes located immediately upstream and downstream of it. (**D**) A phylogenetic tree indicating the 10 defined phylostrata, and a pie chart of fractions of all expressed genes per phylostratum. (**E**) Coefficient of variance of gene expression across the *S. arctica* life cycle, binned by phylostratum. (**F**) Transcriptome age index (TAI) of gene expression for the *S. arctica* life cycle. Higher TAI represents a younger transcriptome.

The online version of this article includes the following source data and figure supplement(s) for figure 4:

**Source data 1.** Table of Blast e-values of orthologs of *S. arctica* lincRNAs in other ichthyosporean species.
**Source data 2.** List of transcriptomes used for lincRNA conservation analysis.
**Source data 3.** Table of gene orthogroups.
**Source data 4.** List of transcriptomes used for generation of orthogroups and phylostratigraphy analysis.
**Source data 5.** Table of *S. arctica* transcripts per gene stratum and orthogroup membership.
**Figure supplement 1.** Long intergenic non-coding RNA (lincRNAs) conservation.
**Figure supplement 2.** Gene expression by phylostrata analysis.

find that the cellularization clusters were enriched for younger genes (*Figure 4—figure supplement 2B and C*). Importantly, we found that genes with ichthyosporean origins were significantly enriched in all three of the largest cellularization clusters (*Figure 4—figure supplement 2B and C*). Finally, computing the transcriptome age index (*Domazet-Lošo and Tautz, 2010*) to assess the overall age of the transcriptome across the life cycle revealed a statistically significant hourglass pattern (p=4.4 * $10^{-5}$ and p=0.02 for replicates 1 and 2, Reductive Hourglass test; *Drost et al., 2015*), with older genes expressed at later stages of growth, and younger genes expressed during early growth and cellularization (*Figure 4F*). Such an hourglass transcriptomic pattern has previously been observed in

animal development, where it reflects the morphological similarities and differences of embryos of different taxa (*Domazet-Lošo and Tautz, 2010*), and it has been suggested as a conserved logic of embryogenesis across kingdoms (*Quint et al., 2012*). Despite this, we currently do not have a morphological explanation for this transcriptional hourglass pattern in ichthyosporeans. Altogether, the phylostratigraphic analysis suggests that cellularization is a comparatively younger process, whereas the growth stage represent an evolutionarily ancient process.

## Temporal co-expression of actin cytoskeleton, cell adhesion and cell polarity pathways during cellularization

To functionally assess the gene expression clusters, we also carried out gene ontology (GO) enrichment analysis (*Figure 5—source data 1*). The largest growth cluster (cluster 1) was enriched in GO terms related to cell growth and biosynthesis. Early and mid-cellularization clusters 5 and 6 were enriched for GO terms related to membrane organization and actin cytoskeleton. Late cellularization clusters were, in addition to GO terms related to actin, also enriched for GO terms related to cell adhesion and polarity. Given that these processes play a major role during cellularization of the insect blastoderm, we investigated the expression pattern of homologs of known regulators of cellularization in *Drosophila*.

In the *Drosophila* blastoderm, cellularization is regulated by several zygotically transcribed genes (*Mazumdar and Mazumdar, 2002*) and relies on extensive membrane trafficking controlled by Rab GTPases (*Bucci et al., 1992*; *Dollar et al., 2002*; *Pelissier et al., 2003*). It also depends on the spatial organization of both the microtubule and actin cytoskeleton. It involves several microtubule and actin binding proteins, including kinesins, Myosin II , Myosin V, profilin (Chickadee), cofilin (Twinstar), formin (Diaphanous) and Septins, and the conserved family of Rho GTPases (*Mazumdar and Mazumdar, 2002*).

In *S. arctica,* we did not find homologs of any *Drosophila* zygotically transcribed genes known to regulate cellularization (data not shown). However, we found that the expression of homologs of Rab5 and Rab11 as well as tubulins and kinesins, except for one (*S. arctica* Kinesin 2), is constant throughout the coenocytic cycle (*Figure 5A and B*, *Figure 5—figure supplement 1*). On the other hand, we found many actin-associated genes dynamically expressed. All actin nucleators of the formin family and members of the Arp2/3 complex peaked during cellularization (*Figure 5C*). In contrast to formins, which largely exhibited sharp peaks, the gene expression of the Arp2/3 complex was initiated earlier and increased gradually (*Figure 5C*). Septins, cofilin, profilin and myosin V were temporally co-expressed during mid-cellularization and peaked at the same time as actin nucleators (*Figure 5D*), whereas myosin II, which has a role in organizing actin filaments and contractility, peaked later (*Figure 5D*). We likewise found the expression of three out of four members of the Cdc42/Rho1 orthogroup to be sharply activated during late cellularization (*Figure 5E*). Temporal co-expression of these genes suggests that the cellular pathways responsible for organizing the cytoskeleton and cell polarity in *Drosophila* cellularization are also involved in the cellularization process of *S. arctica.*

Since GO enrichment analysis suggested expression of cell adhesion genes, we investigated expression pattern of conserved cell-cell and cell-matrix adhesion pathways. Integrins and the cytoplasmic members of the integrin adhesome mediate cell-matrix adhesion in animal tissues. We observed that in *S. arctica*, both integrin receptors, alpha and beta, as well as all cytoplasmic members, are temporally co-expressed during late cellularization (*Figure 5F*). Beta and alpha catenin, together with cadherins in animals, mediate cell-cell adhesion in epithelial tissues (*Rodriguez-Boulan and Macara, 2014*). In *S. arctica*, we found three copies of Aardvark, a homolog of beta-catenin (*Murray and Zaidel-Bar, 2014*), as well as one homolog of alpha-catenin (*Miller et al., 2013*). We found that the expression of two out of three Aardvark transcripts, as well as the expression of the alpha catenin homolog, peaked during late cellularization (*Figure 5F*). This suggests that both cell-matrix and cell-cell adhesion pathways, largely conserved between animals and ichthyosporeans, play a role in the establishment of the polarized layer of cells during late cellularization.

## The actomyosin cytoskeleton is essential for cellularization

Finally, we tested whether disrupting the cytoskeleton would lead to defects in cellularization. In the absence of genetic tools, we used small inhibitory molecules that target specific conserved

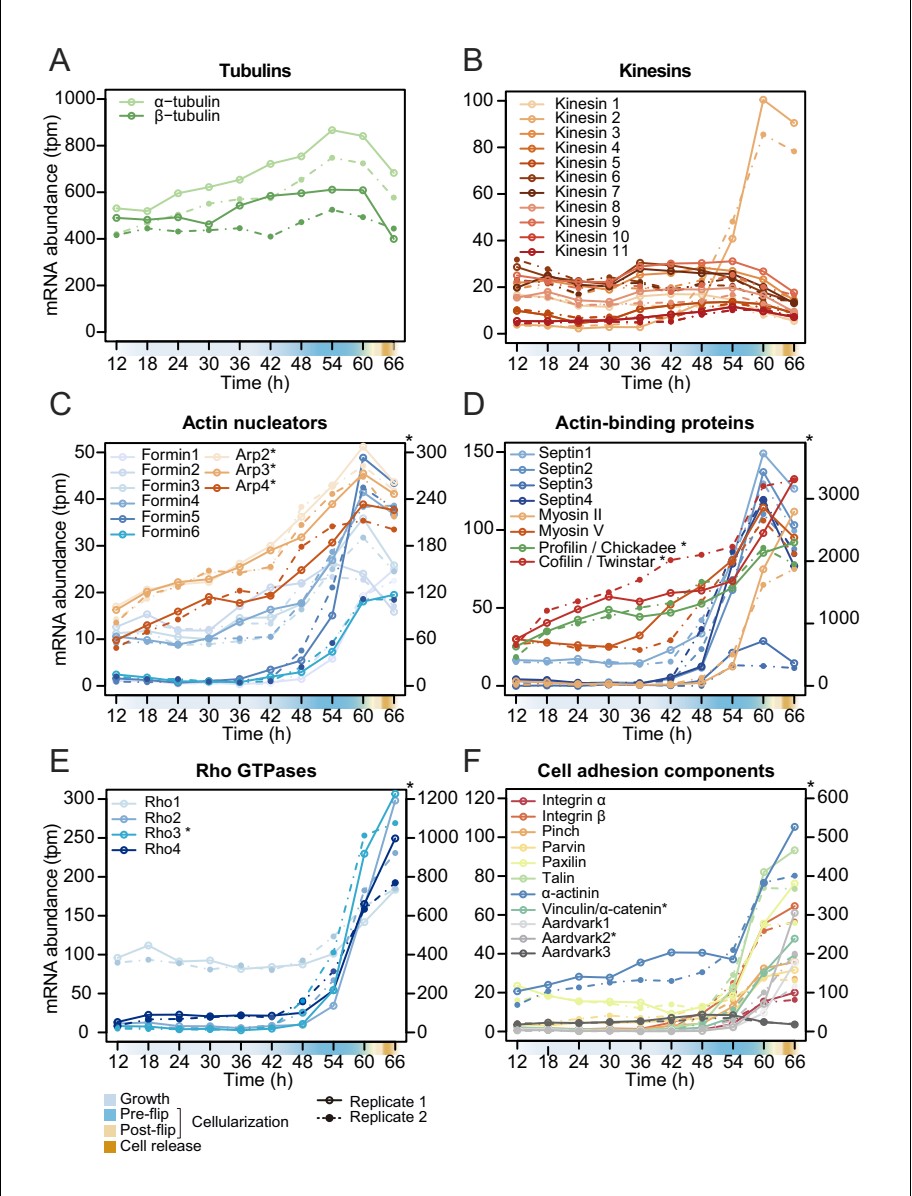

**Figure 5.** Temporal transcript abundance of cytoskeletal, cell polarity and cell adhesion genes. (**A–F**) Gene expression of indicated genes across the *S. arctica* life cycle.

The online version of this article includes the following source data and figure supplement(s) for figure 5:

**Source data 1.** Gene ontology (GO) enrichment analysis of gene expression clusters.

**Source data 2.** A table of reference gene names plotted in *Figure 5*.

**Figure supplement 1.** Gene expression of Rab5 and Rab8 homologs across the *S. arctica* life cycle.

cytoskeletal components. We synchronized the cultures and added the inhibitors at the onset of cellularization (54 hr time point). We first assessed the role of the microtubule cytoskeleton during cellularization by adding carbendazim (MBC), a microtubule depolymerizing agent (*Castagnetti et al., 2007*). Microtubule inhibition did not prevent plasma membrane invagination, although it resulted in a delay of flip and release of newborn cells (*Figure 6A and B*, and *Figure 6—figure supplement 1A*, *Videos 6* and *7*).

Furthermore, by staining the MBC-treated cells with DAPI and phalloidin, we observed a loss of the uniform spacing of the nuclei and actin filaments during the cortical compartmentalization stage of cellularization (*Figure 6C*). After MBC treatment, newborn cells varied in size and number of

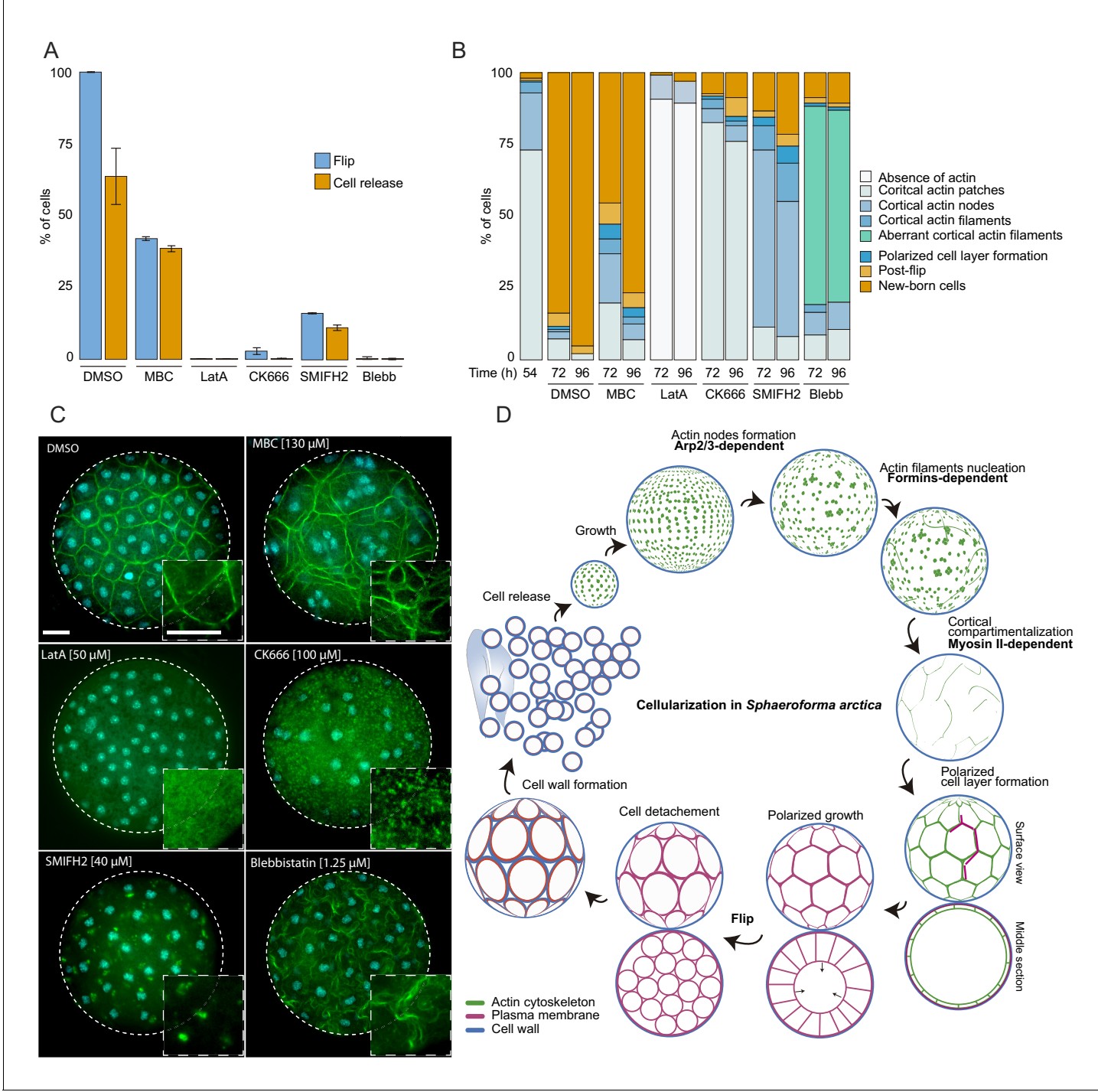

**Figure 6.** The actomyosin network organization is crucial for cellularization of *S. arctica*. (**A**) Depolymerization of microtubules and actin, as well as inhibition of Arp2/3, formins and myosin II affect 'flip' and release of newborn cells. Synchronized cells of *S. arctica*, pre-grown for 54 hr, were imaged for 24 hr in presence of multiple pharmacological inhibitors (DMSO as a control, MBC, Latrunculin A, CK666, SMIFH2, Blebbistatin). Cells undergoing flip and cell release throughout the duration of the experiment were measured (N > 400 cells, Error bars are standard deviation from three independent experiments). (**B**) Temporal functions of the Arp2/3 complex, formins and myosin II in distinct stages of cellularization. Synchronized cells of *S. arctica*, pre-grown for 54 hr were subject to multiple pharmacological inhibitors treatments (DMSO as a control, MBC, Latrunculin A, CK666, SMIFH2, Blebbistatin) and fixed and stained with phalloidin and DAPI after 24 hr and 48 hr of treatments. Phalloidin staining allows us to measure the fraction of cells exhibiting different actin and cellular structures throughout cellularization. (**C**) The different actin structures observed when cells are treated with multiple pharmacological inhibitors treatments (DMSO as a control, MBC, Latrunculin A, CK666, SMIFH2, Blebbistatin). Bar, 10 μm. (**D**) A model

*Figure 6 continued on next page*

Figure 6 continued

representing the actin cytoskeleton, plasma membrane and cell wall at different stages of the cellularization process in *S. arctica*, indicating sequential steps of actin remodeling mediated by Arp2/3, formins and Myosin II.

The online version of this article includes the following figure supplement(s) for figure 6:

**Figure supplement 1.** Inhibition of the actin cytoskeleton blocks plasma membrane invaginations.

nuclei (*Figure 6—figure supplement 1B and C*, *Video 6*). These results suggest that the microtubule cytoskeleton is not essential for plasma membrane invagination but is crucial for nucleus and actin filament positioning at the cortex of the coenocyte. This is in contrast to the role of microtubules during cellularization of the blastoderm in *Drosophila*, where microtubules also directly drive the plasma membrane invagination through formation of inverted baskets covering the nuclei (*Mazumdar and Mazumdar, 2002*).

Next, we sought to disrupt actin polymerization using either the broad actin depolymerizing agent Latrunculin A (LatA) (*Braet et al., 1996*), the Arp2/3 inhibitor CK666 (*Hetrick et al., 2013*), or the formin inhibitor SMIFH2 (*Kim et al., 2015b*). Cells treated with Latrunculin A lacked any actin patches or actin filaments and failed to undergo flip or produce newborn cells (*Figure 6A-C*, and *Figure 6—figure supplement 1A*, *Videos 6* and *7*). Furthermore, plasma membrane invagination did not occur after LatA treatment (*Videos 6* and *7*). In contrast, CK666-treated cells formed cortical actin patches, but were not able to form actin nodes, and they were unable to generate plasma membrane invaginations (*Figure 6A–6C*, *Videos 6* and *7*). These results show that the actin cytoskeleton and Arp2/3-mediated actin nucleation are required for the formation of actin nodes, the first step in the cellularization of *S. arctica*. In contrast, the formin inhibitor SMIFH2 did not block the formation of actin nodes, but it prevented the formation of actin filaments in the later stages (*Figure 6A–6C*). In addition, the plasma membrane invagination did not occur (*Figure 6A-C*, and *Figure 6—figure supplement 1A*, *Videos 6* and *7*), although we note that a small fraction of cells was not affected. This suggests that formins play a role in the nucleation of actin filaments after the formation of actin nodes.

Finally, we assessed the role of Myosin II, using blebbistatin, an inhibitor of Myosin II ATPase activity (*Kovács et al., 2004*). Blebbistatin treatment blocked plasma membrane invaginations and prevented cellularization (*Figure 6A-C*, and *Figure 6—figure supplement 1A*, *Videos 6* and *7*). In coenocytes where plasma membrane invaginations started but the cell layer was not yet formed, we observed a retraction of already present invaginations when treated with blebbistatin (*Figure 6—figure supplement 1C*, *Video 8*). Additionally, in coenocytes where the cell layer was formed but polarized growth was not complete, blebbistatin treatment prevented polarized growth but allowed the release of cells of different cell sizes (*Figure 6—figure supplement 1C*, *Video 8*). Although blebbistatin-treated cells were able to form actin filaments, they had an aberrant wavy cortical actin filament network, suggesting that inhibition of Myosin II causes loss of actin crosslinking and actin network contractility (*Figure 6C*). Taken together, these results indicate that the actomyosin apparatus is essential for cellularization in *S. arctica* and reveal a temporal sequence of involvement of Arp2/3 complex, formins and Myosin II (*Figure 6D*). This temporal sequence is reflected in the relative timing of expression of Arp2/3, formins and myosin II genes (*Figure 5C and D*).

## Discussion

To address whether animal-like cellularization exists outside animals, we performed imaging, transcriptomic analysis and pharmacological inhibition experiments in an ichthyosporean *Sphaeroforma arctica*, a unicellular relative of animals (*Figure 6D*). We show that at the onset of cellularization, Arp2/3 complex mediates the formation of actin nodes at the cortex of the coenocyte. This is followed by formin-dependent nucleation of actin filaments. These actin

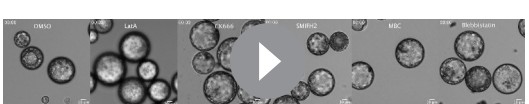

**Video 6.** Time lapse of synchronized cells of *S. arctica*, pre-grown for 54 hr and treated with different pharmacological inhibitors. Time interval between frames is 30 min. The movie is played at 7fps. Cellularization is affected when cells are treated with LatA, CK666, SMIFH2 and Blebbistatin. Bar, 10 μm.
https://elifesciences.org/articles/49801#video6

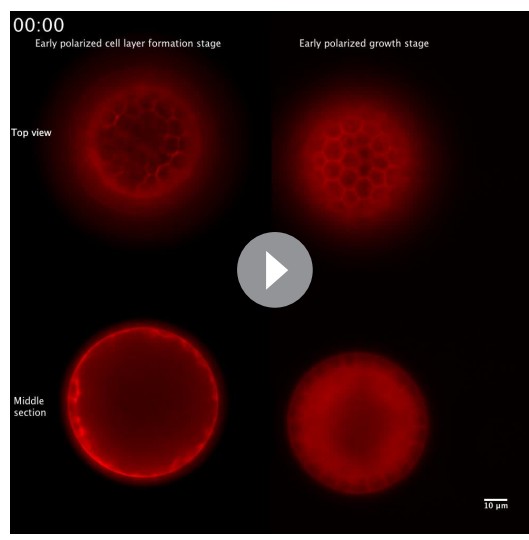

**Video 7.** Time lapse of synchronized cells of *S. arctica*, pre-grown for 54 hr, stained with the plasma membrane dye FM4-64 and treated with different pharmacological inhibitors. Time interval is 10 min. The movie is played at 7fps. Plasma membrane invaginations is prevented when cells are treated with LatA, CK666, SMIFH2 and Blebbistatin. Bar, 10 μm.
https://elifesciences.org/articles/49801#video7

**Video 8.** Time lapse of synchronized cells of *S. arctica*, pre-grown for 62 hr, stained with the plasma membrane dye FM4-64 and treated with blebbistatin. Time interval is 5 min. The movie is played at 7fps. Plasma membrane invaginations and polarized growth are affectd when cells are treated with Blebbistatin. Bar, 10 μm.
https://elifesciences.org/articles/49801#video8

filaments are then crosslinked in a Myosin II-dependent manner, which results in formation of a cortical actomyosin network that surrounds evenly spaced nuclei. This spatial organization of both the actomyosin network and nuclei depends on microtubules. The localization of the actomyosin network determines the sites of plasma membrane invaginations and drives their inward growth through a slow and a fast phase, resulting in the formation of a polarized layer of cells. This stage co-occurs with transcriptional activation of genes involved in cell-cell and cell-matrix adhesion. Polarized growth persists until the internal cavity is occupied, after which the polyhedrally-shaped cells undergo flip, lose polarization and detach from each other. At this point, future newborn cells form the cell wall before they are released and initiate a new coenocytic cycle. Overall, our results show that the cellularization of *S. arctica* morphologically resembles the cellularization of the *Drosophila* blastoderm and share a common actomyosin pathway for cellularization. This suggests two possible evolutionary scenarios. First, given the widespread presence of coenocytes in opisthokonts, we have suggested that the common ancestor of opisthokonts already had a coenocytic life cycle (*Ondracka et al., 2018*). In this scenario, cellularization of the coenocyte by this conserved actomyosin mechanism would have already been present in this common ancestor. In the second scenario, coenocytes in animals and ichthyosporeans would have evolved independently, but adopted the same conserved actomyosin mechanism for cellularization.

Additionally, our transcriptomic analysis suggests that transcriptional control plays an important role in regulation of the coenocytic cycle of *S. arctica*. Interestingly, we show that many genes expressed during cellularization of *S. arctica* emerged at the onset of ichthyosporeans. This strongly suggests that the general mechanisms of cellularization in *S. arctica* are likely conserved within ichthyosporeans. Altogether, our results argue that cellularization in ichthyosporeans requires both regulatory pathways conserved with animals, as well as regulatory pathways that evolved independently in ichthyosporeans. The later may include lincRNAs that we found to be remarkably conserved among ichthyosporeans and show high expression levels. Overall, our temporal gene expression dataset provides a good resource for further functional studies in *S. arctica* and other ichthyosporean species.

Among pathways transcriptionally upregulated during cellularization are the conserved cell-cell adhesion and cell-matrix adhesion pathways, which occur concomitantly with the formation of polarized layer of cells. These pathways are hallmarks of epithelial tissues in animals (*Rodriguez-Boulan and Macara, 2014*). The epithelium is the first tissue that is established during embryogenesis and perhaps the first that emerged in evolution, thus representing the basic form of multicellular organization in animals (*Brunet and King, 2017*). Epithelia are found in all animal lineages, including sponges (*Leys et al., 2009*). It has previously been suggested that the origin of epithelial structures predates the origin of animals, as a polarized non-cadherin-based epithelium-like structure,

regulated by alpha and beta catenins, is also present in the slime mold *Dictyostelium discoideum* (***Dickinson et al., 2011***). Based on this indirect evidence from transcriptomics, we hypothesize that cellularization in *S. arctica* generates a transient epithelium-like layer of cells, lending support to the hypothesis that the origin of epithelial tissues predates the origin of animals (***Dickinson et al., 2012***). However, unlike the epithelium formation in social amoebae, which originates through aggregation (***Bonner, 1998***), ichthyosporean epithelium is generated clonally, such as in animals. Further comparison of this ichthyosporean epithelium-like structure with animal epithelia will allow us to address whether *S. arctica* form a bona fide epithelium during cellularization.

# Materials and methods

**Key resources table**

| Reagent type (species) or resource | Designation | Source or reference | Identifiers | Additional information |
|---|---|---|---|---|
| Commercial assay or kit | miRNeasy Mini Kit | QIAGEN | Cat. #: 217004 | |
| Commercial assay or kit | RNeasy Mini Kit | QIAGEN | Cat. #: 74104 | |
| Commercial assay or kit | QIAamp DNA Blood Midi Kit | QIAGEN | Cat. #: 51183 | |
| Chemical compound, drug | FM4-64 | Invitrogen | Cat. #: T3166 | |
| Chemical compound, drug | FM4-64FX | Invitrogen | Cat. #: F34653 | |
| Chemical compound, drug | Latrunculin A | Sigma-Aldrich | Cat. #: L5163 | |
| Chemical compound, drug | CK666 | Sigma-Aldrich | Cat. #: SML0006 | |
| Chemical compound, drug | SMIFH2 | Sigma-Aldrich | Cat. #: S4826 | |
| Chemical compound, drug | Blebbistatin | Sigma-Aldrich | Cat. #: B0560 | |
| Chemical compound, drug | Carbendazim (MBC) | Sigma-Aldrich | Cat. #: 378674 | |
| Software | ImageJ (http://imagej.nih.gov/ij/) | PMID: 29187165 | | |
| Software | FALCON_ unzip assembler | | | v0.4.0 |
| Software, algorithm | SMRT analysis | | | v2.3.0 |
| Software, algorithm | BWA mem | | | v0.7.12 |
| Software, algorithm | Pilon | PMID: 25409509 | | v1.22 |
| Software, algorithm | BRAKER2 (https://github.com/Gaius-Augustus/BRAKER) | PMID: 26559507 | | |
| Software, algorithm | STAR 2.7 | PMID: 23104886 | | |
| Software, algorithm | Augustus 3.2.2 | PMID: 21216780 | | |
| Software, algorithm | Genemark v4.21 | PMID: 16314312 | | |
| Software, algorithm | Trimgalore https://github.com/FelixKrueger/TrimGalore | | | v0.4.5 |

*Continued on next page*

*Continued*

| Reagent type (species) or resource | Designation | Source or reference | Identifiers | Additional information |
|---|---|---|---|---|
| Software, algorithm | Trimmomatic | PMID: 24695404 | | v0.35 |
| Software, algorithm | Trinity | PMID: 21572440 | | v2.5.1 |
| Software, algorithm | Hisat2 | PMID: 25751142 | | v2.1.0 |
| Software, algorithm | PASA pipeline | PMID: 14500829 | | v2.3.1 |
| Software, algorithm | Kallisto software | PMID: 27043002 | | |
| Software, algorithm | Clust software | PMID: 30359297 | | |
| Software, algorithm | eggNOG mapper | PMID: 28460117 | | |
| Software, algorithm | Spades | PMID: 24093227 | | v3.10.0 |
| Software, algorithm | Count software | PMID: 20551134 | | |

## Culture conditions

*Sphaeroforma arctica* cultures were grown and synchronized as described previously (*Ondracka et al., 2018*). Briefly, cultures were grown in Marine Broth (Difco BD, NJ, USA; 37.4 g/L) in flasks at 12°C until saturation, producing small cells. These cells were then diluted into fresh media with low density (1:300 dilution of the saturated culture), resulting in a synchronously growing culture. Cultures of *S. gastrica*, *S. tapetis*, *S. nootkatensis*, *P. gemmata* and *A. whisleri* were similarly kept in Marine Broth (Difco BD, NJ, USA; 37.4 g/L) at 12°C in dark conditions.

## *Sphaeroforma arctica* genome sequencing and assembly

Genomic DNA from *S. arctica* was extracted using QIAamp DNA Blood Midi Kit (Qiagen) from 300 mL culture incubated at 12 C° for 1 week in six 75 $cm^2$ flasks. The Qubit (Invitrogen) quantification found ~150 µg genomic DNA in total.

A SMRTbell library for P6/C4 chemistry was constructed and was run on 32 SMRT cells in a PacBio RSII system (Pacific Biosciences), generating 2,209,004 subreads with a total of 25,164,714,269 bp. Raw subreads were first assembled using the FALCON_unzip assembler (v0.4.0) and the initial assembled sequences were polished by the Quiver integrated in SMRT analysis (v2.3.0). Genomic DNA was then sheared using a Focused-ultrasonicator (Covaris, Inc). A paired-end library with an insert size of 600 bp was sequenced on an Illumina HiSeq2500 platform, producing 136,566,600 reads with read length of 250 bp. Paired-end reads were mapped against the polished sequences using the BWA mem (v0.7.12) followed by error-correction using Pilon (v1.22) (*Walker et al., 2014*).

The assembled genome is deposited on DDBJ (DNA Data Bank of Japan) under accession numbers BJTW01000001-BJTW01000733.

## *Sphaeroforma arctica* gene annotation

We annotated predicted gene models in our scaffolds using BRAKER2 (*Hoff et al., 2018*). First we used STAR 2.7 (*Dobin et al., 2013*) to map all the RNA-seq samples to the genomic scaffolds, so as to obtain empirical evidence of gene bodies and guide the prediction of gene coordinates. These read mappings were then supplied to BRAKER2 in BAM format, which combied these external evidence with the gene coordinates predicted by Augustus 3.2.2 (*Keller et al., 2011*) (–noInFrameStop mode) and Genemark v4.21 (*Lomsadze et al., 2005*) to obtain a consolidated set of gene models in GFF format.

The transcriptome of *S. arctica* was assembled in order to add 5' and 3' UTR regions to the genome annotation and to search for long non-coding RNA-seq using data obtained here (see below).

The RNA-seq reads from the first time series replicate (excluding timepoint 30) were subjected to quality trimming and adaptor removal using Trimgalore (v0.4.5) (http://www.bioinformatics.babra-ham.ac.uk/projects/trim_galore/) and Trimmomatic v0.35 (*Bolger et al., 2014*). Trimgalore was first run to remove sequencing adaptors, allowing 0 mismatch and minimum adaptor length of 1 nt. Trim-momatic was run by trimming bases with phred score <20 from both ends. Furthermore, a sliding window of 4 bases was used to trim reads from the 5' end when the mean phred score dropped below 20. Finally, the IlluminaClip option was used to search for remaining sequencing adaptors, allowing two seed mismatches, a palindrome clip threshold of 20 and minimum single match thresh-old of 7 nt. The quality filtered RNA reads were subsequently assembled using Trinity v2.5.1 (*Grabherr et al., 2011*), running both a de novo and a genome-guided assembly. To perform a genome-guided transcriptome assembly, we first mapped the RNA-seq reads against the genome using Hisat2 v2.1.0 (*Kim et al., 2015a*), in strand-specific mode with default parameters. Both Trinity assemblies were run in strand-specific mode while applying the jaccard clip option and otherwise default parameters. We next evaluated the strand-specificity of the assemblies by mapping RNA-seq reads back to the Trinity assemblies using scripts supplied within the Trinity software. Transcripts were subsequently removed if >80% of the reads mapped in the wrong direction. The PASA pipeline v2.3.1 (*Haas et al., 2003*) was then used to update the existing genome annotation. First, the assembled transcripts described above were mapped against the genome using the initial PASA script to make a temporary annotation file. This was performed with default parameters, using both Blat v35 (*Kent, 2002*) and Gmap v2015-09-29 (*Wu and Watanabe, 2005*) aligners, with transcripts specified as strand-specific. Lastly, the transcriptome-based annotations were compared with the existing genome annotation in a final PASA run, also using default parameters. In this step, the exist-ing genes were expanded with UTR annotations and, in cases where a single RNA transcript covered multiple genes, these become merged into a single gene. The annotated genome and proteome of *S. arctica* can be found on figshare; https://figshare.com/authors/Multicellgenome_Lab/2628379.

## RNA isolation, library preparation and sequencing

Synchronized cultures of *Sphaeroforma arctica* at 12°C were sampled every 6 hr for a total duration of 72 hr. Total RNA was extracted by Trizol and purified using the miRNeasy Mini Kit (QIAGEN) from ~50 mL of culture at each time point. Libraries were prepared using the TruSeq Stranded mRNA Sample Prep kit. Paired-end 50 bp read length sequencing was carried out at the CRG geno-mics core unit on an Illumina HiSeq v4 sequencer. We obtained between 19 and 32 M reads per sample. Transcripts were quantified using Kallisto software (*Bray et al., 2016*) with default parame-ters. To remove non-expressed genes, we filtered out transcripts that had a mean expression level of <0.5 tpm across all 20 samples. This resulted in a set of 14557 transcripts that were used for clustering.

For *S. gastrica* and *S. tapetis*, RNA purification was performed using the RNeasy kit (Qiagen), while for *S. nootkatensis*, *P. gemmata* and *A. whisleri*, RNA was purified using Trizol (Invitrogen, CA, USA). Strand-specific sequencing libraries were prepared using Illumina TrueSeq Stranded mRNA Sample Prep kit and sequenced on an Illumina HiSeq3000 machine (150 bp paired end).

## Clustering of the gene expression and gene ontology enrichment analysis in *S. arctica*

*Sphaeroforma arctica* transcripts were clustered by their gene expression profiles using Clust soft-ware (*Abu-Jamous and Kelly, 2018*) with default parameters and automatic normalization mode. This yielded 4511 transcripts (4441 coding genes and 70 lncRNA genes) that were clustered into nine clusters, ranging from 41 to 2081 co-expressed genes. Gene expression profiles were visualized using superheat package in R (*Barter and Yu, 2018*). The tSNE plot was generated using Rtsne package. The code used for the transcriptome analysis and visualization is available from GitHub: https://github.com/andrejondracka/sphaeroforma_transcriptome (*Ondracka, 2019*; https://github.com/elifesciences-publications/sphaeroforma_transcriptome).

Gene Ontology enrichments based on the GOs annotated with eggNOG mapper (*Huerta-Cepas et al., 2017*) were computed using the topGO R library (v. 2.34). Specifically, we computed the functional enrichments based on the counts of genes belonging to the group of interest relative to all annotated genes, using Fisher's exact test and the elim algorithm for ontology weighting (*Alexa et al., 2006*).

## Transcriptome assembly of other unicellular holozoans

Raw reads (from sequencing libraries and SRA data) were processed with Trimmomatic (*Bolger et al., 2014*) to remove adapters and low-quality bases, by trimming bases with phred score <28 from both ends. Furthermore, a sliding window of 4 bases was used to trim reads from the 5' end when the mean phred score dropped below 28. Finally, the IlluminaClip option was used to search for remaining sequencing adaptors, allowing two seed mismatches, a palindrome clip threshold of 28 and minimum single match threshold of 10 nt. All libraries were assembled denovo with Trinity (v2.3.2–2.5.1) (*Grabherr et al., 2011*) using default parameters. The assemblies of RNA-seq data were performed with the strand specific option, while assemblies based on SRA data were run in standard mode. For the *S. sirkka* assembly we also applied the jaccard clip option. Coding regions were predicted using TransDecoder v5.2.0 (*Haas et al., 2013*), by first extracting the longest possible ORFs (only top strand was searched in the strand-specific assemblies), on which likely coding region was predicted. Only the longest ORF was kept for each transcript.

## DNA isolation and genome assembly of *S. gastrica* and *S. tapetis*

Genomic DNA (gDNA) was isolated from *S. gastrica* and *S. tapetis*, by lysing cells on a FastPrep system (MP Biomedicals, CA, USA; 4 m/s, 20 s) followed by gDNA purification using the DNeasy kit (Qiagen, NRW, Germany) and subsequently sequenced on an Illumina HiSeq X system (150 bp paired end). Raw reads were subjected to quality trimming and adaptor removal as described above for RNA-seq data, with a phred score cut-off of 26.

The quality trimmed reads were subsequently error-corrected and assembled using Spades v3.10.0 (*Nurk et al., 2013*) applying kmer values 21, 33, 55, 77, 99 and 121, but otherwise default parameters. The resulting spades assemblies were scaffolded using L_rna_scaffolder (*Xue et al., 2013*) and polished with Pilon (*Walker et al., 2014*). L_RNA_Scaffolder was run by first mapping the respective transcriptome assemblies to the genome assemblies using Blat (*Kent, 2002*) which was inputted to L_rna_scaffolder. Next, we run Pilon by first mapping the quality trimmed genomic Illumina reads to the genome assembly using Bowtie2 (*Langmead and Salzberg, 2012*). The resulting mapping file was then used in Pilon with default parameters. L_rna_scaffolder and Pilon were run repeatedly 5 times, followed by three final runs using only Pilon. These genome assemblies were used as reference genomes for genome-guided Trinity assemblies and PASA annotation as previously described for *S. arctica*. The genomic reads for *S. gastrica* and *S. tapetis* can be found under the EBI/ENA accession: PRJEB34306.

## *S. arctica* long intergenic non-coding RNA identification and conservation analysis

Long intergenic non-coding RNAs (lincRNAs) were identified from the PASA annotation described above. First, transcripts shorter than 200 nt were discarded. Then, coding potential was evaluated using both TransDecoder (*Haas et al., 2013*) and CPC2 (*Kang et al., 2017*) with default parameters. All transcripts lacking coding potential were then compared with the Rfam database (*Kalvari et al., 2018*) to exclude other non-coding RNAs such as rRNAs and tRNAs. To exclude possible UTRs actually belonging to fragmented protein coding genes, a minimum genomic distance of 1000 bp from the closest protein coding gene was required. The remaining transcripts were compared with protein coding genes in the Swissprot database using Blastx (*Altschul et al., 1990*), and all sequences with a match smaller than e-value 1e-5 were removed.

Next, the potential lncRNA transcripts were blasted against the proteomes of *S. arctica* and other closely related ichthyosporeans (*Sphaeroforma tapetis*, *Sphaeroforma sirkka*, *Sphaeroforma nootkatensis*, *Sphaeroforma napiecek*, *Sphaeroforma gastrica*, *Creolimax fragrantissima*, *Pirum gemmata* and *Abeoforma whisleri*), and all transcripts with a match smaller than e-value 1e-5, were discarded. To remove transcripts resulting from potential spurious transcription, we required a minimum

expression level of at least one tpm in at least one time series sample. The transcriptome reads for *S. sirkka* and *S. napiecek* can be found under the EBI/ENA accession: ERR2729814 and ERR2729813 respectively. The transcriptome reads for *S. tapetis*, *S nootkatensis* and *S. gastrica* can be found under the EBI/ENA accession: PRJEB34306.

To search for possible conserved lincRNAs in other species, we performed Blastn searches against transcriptomes and genomes of several holozoan species (*Richter et al., 2018*; *de Mendoza et al., 2015*; *Sebé-Pedrós et al., 2013*) (*Figure 4—source data 2*). All transcripts providing hits with an e-value less than 1e-50 were considered to be a conserved homolog.

## Genome-wide analysis of alternative splicing

Each RNA-seq run was independently aligned to the *Sphaeroforma* genome using Hisat2 (*Kim et al., 2015a*), using default parameters except for longer anchor lengths to faciliate de novo transcriptome assembly (–dta flag).

The resulting alignments (in SAM format) were used to build sample-specific transcriptome assemblies with Stringtie2, using existing gene models (in GFF format, -G flag) as a reference, a minimum isoform abundance of 0.01 (-f flag) and a minimum isoform length of 50 bp (-m flag). For each sample, we only retained transcripts that overlapped with known genes in the final GFF file (using bedtools; *Quinlan and Hall, 2010*). Then, we built a consolidated set of isoforms by pooling all sample-specific GFF annotations and the reference annotation using Stringtie2 (–merge flag), without imposing any limitation of minimum expression levels (-T flag set to 0), and retaining isoforms with retained introns (-i flag). We also calculated the expression levels at the isoform level using Salmon (*Patro et al., 2017*) (output in TPM). Then, we used SUPPPA2 (*Trincado et al., 2018*) to generate a set of alternative splicing events, for which their frequencies were calculated for each sample. Specifically, we used the consolidated set of isoforms (GFF format) to obtain a list of all possible AS events using SUPPA2 *generateEvents* mode (setting the output to exon skipping [SE], mutually exclusive exons [MX] and intron retention [RI]), and -l 10. Then, we used SUPPA psiPerEvent mode to calculate the PSI values of each AS event for each sample, using the expression levels of each isoform (obtained from Salmon) as a reference. Differential splicing was quantified by calculating the calculating the differential PSI values between the average of each sample group (growth stage [t = 12 hr to t = 48 hr] compared to cellularization stage [t = 54 hr to t = 66 hr]). *p*-values were obtained using the empirical significance calculation protocol described in SUPPA2.

After running SUPPA2, we produced functional annotations of the effect of each AS event on the final transcript. First, we annotated the protein domains in our consesus gene models using Pfam (version 31) and Pfamscan. These protein-level coordinates were converted to genomic coordinates using BLAT alignments (version 36, by aligning protein sequences to 6-frame translations of the genome) (*Kent, 2002*) and intersected with the genomic coordinates of AS events using the GenomicRanges and IRanges libraries (*findOverlapPairs* module) (*Lawrence et al., 2013*) from the R statistical framework (version 3.5.2). Second, we used the genomic coordinates of all AS events to recode each AS event as insertion/deletion variants that were amendable for analysis using the Variant Effect Predictor software (version 92.3) (*McLaren et al., 2016*) (SE were encoded as deletions, RI as insertions, and MX as insertion/deletion complex events). VEP produced a list of the effects of each AS variant (according to the Sequence Ontology nomenclature; *Eilbeck et al., 2005*), using our consensus gene models as a reference.

## Phylostratigraphy analysis

First, we performed gene orthology assignment by searching for orthologs in a representative set of 30 publicly available eukaryotic proteome sequences (animals: *H. sapiens*, *S. kowalewskii*, *D. melanogaster*, *T. tribolium*, *N. vectensis*, *T. adherens*, *M. leidii* and *A. queenslandica*; choanoflagellates: *S. rosetta*; filastereans: *C. owczarczaki* and *M. vibrans*; teretosporeans: *S. arctica*, *C. fragrantissima*, *I. Hoferi*, *A. whisleri*, *C. perkinsii*, *P. gemmata*, *S. destruens*, *C. limatocisporum*, fungi: *F. alba*, *C. anguillulae*, *S. punctatus*, *M. verticillata* and *S. pombe*; other eukaryotes: *T. trahens*, *A. castellanii*, *D. discoideum*, *N. gruberi*, *T. thermophila*, *E. huxleyi*, *A. thaliana* and *P. yezoensis*) using orthofinder (*Emms and Kelly, 2015*) in Diamond mode. The longest isoform of each transcript was used. The generated orthogroups were used to determine orthologs of *S. arctica* genes in other species, unless a phylogeny of a specific gene family has been published.

The orthofinder analysis resulted in 18797 orthogroups. To determine gene age of each orthogroup, the orthofinder output was passed to Count software (*Csurös, 2010*) to classify the *S. arctica* proteome into 10 phylostrata according to Dollo parsimony, ranging from paneukaryotic to *S. arctica*-specific. *S. arctica* genes that were not present in any orthogroup (i.e. did not have a single ortholog in any of the species) were also classified as *S. arctica*-specific. The enrichment analyses were performed using Fisher test implemented in R.

The transcriptome age index (TAI) at each time point was computed as:

TAI = sum$_i$(ps$_i$ * e$_i$)/sum$_i$(e$_i$) where ps$_i$ is an integer that represent the phylostratum of the gene *i* (with older genes assigned lower ps; for instance one for genes with paneukaryotic origin and 10 for genes with origin in *S. arctica*), and e$_i$ the expression level of each gene *i* (in tpm), according to *Domazet-Lošo and Tautz (2010)*. The reductive hourglass statistical test for the hourglass pattern was implemented using the R package myTAI (*Drost et al., 2015*), with time points 1 and 2 defined as early, time points 3–6 as mid, and time points 7–10 as late phases.

## Flow cytometry

Flow cytometry was performed as described previously (*Ondracka et al., 2018*) Cells were fixed in 4% formaldehyde, 1M sorbitol solution for 15 min at room temperature, washed once with marine PBS (PBS with 35 g/L NaCl), and stained with DAPI (final concentration 0.5 µg/mL) in marine PBS. Samples were analyzed using an LSRII flow cytometer (BD Biosciences, USA) and the data were collected with FACSDiva software. DAPI signal was measured using a 355 nm laser with the 505 nm longpass and 530/30 nm bandpass filters. Approximately 2000 events were recorded in each measurement. The flow cytometry data were processed and analyzed using FloJo software (Ashland, OR).

## Microscopy

Confocal microscopy of the spatiotemporal organization of actin in *Figure 2A* and *Video 5*, was performed using a confocal laser scanning Leica TCS SP5 II microscope with an HC PL APO 63x/1.40 Oil CS2 oil objective. All remaining live and fixed images were obtained using a Zeiss Axio Observer Z.1 Epifluorescence inverted microscope equipped with Colibri LED illumination system and an Axiocam 503 mono camera. A Plan-Apochromat 63X/1.4 oil objective has been used for imaging fixed cells (*Figure 2C*, *Figure 2—figure supplements 1A*; *Figure 6B-D*), and an EC Plan-Neofluar 40x/0.75 air objective for *Figure 2—figure supplement 1C*, *Videos 3* and *4* and an N-ACHROPLAN 20x/0.45na Ph2 air objective for live imaging in *Figure 1C and B*, *Figure 1—figure supplement 1B*, *Videos 1*, *2*, *6*, *7* and *8*.

## Cell fixation and staining

Cells were fixed using 4% formaldehyde and 250 mM sorbitol for 30 min before being washed twice with PBS. For actin and nuclei staining phalloidin (*Figures 2A, C* and *6C*, *Figure 2—figure supplement 1B and C*), cells were span down at 1,000 rpm for 3 min after fixation and washed again three times with PBS before adding 10 µl of Alexa Fluor 488–phalloidin (Invitrogen) and DAPI at a final concentration of 5 µg/mL to 5 µl of concentrated sample. For plasma membrane and cell wall staining (*Figure 2—figure supplement 1D*), cells were incubated for 10 min with FM4-64FX (Invitrogen) at a final concentration of 10 µM from 100 × DMSO diluted stock solution and Calcofluor white (Sigma-Aldrich) at a final concentration of 5 µg/ml from a 200 × stock solution prior to fixation. Cells were then fixed as previously mentioned and concentrated before being disposed between slide and coverslip.

For *Figure 6A*, *Figure 6—figure supplement 1A and B*, cells were pre-grown at 12°C for 48 hr prior to fixing every hour for a total of 14 hr.

## Live microscopy and pharmacological inhibitors

For live-cell imaging (*Figures 1C–1F*, *2B* and *6A*, *Figure 1—figure supplement 1A to C*, *Videos 1– 4* and *6–8*) saturated culture was diluted 300x in fresh marine broth medium 1X inside a µ-Slide 4 or eight well slide (Ibidi) at time zero. To ensure oxygenation during the whole period of the experiment, the cover has been removed. To maintain the temperature at 12°C we used a P-Lab Tek (Pecon GmbH) Heating/Cooling system connected to a Lauda Ecoline E100 circulating water bath.

To reduce light toxicity, we used a 495 nm Long Pass Filter (FGL495M- ThorLabs). Kymographs of cells undergoing cellularization in *Figure 1D* were constructed in ImageJ v1.46 by drawing a 3-pixel-wide (0.39 µm) line crossing the center of each cell.

For plasma membrane live staining (*Figure 2B*, *Figure 2—figure supplement 1A*, *Videos 3*, *4*, *7* and *8*), FM4-64 (Invitrogen) at a final concentration of 10 µM from 100 × DMSO diluted stock solution was added after 58 hr of growth unless indicated otherwise in figure legends.

Treatment with pharmacological inhibitors was performed on 54 hr grown cells inside a µ-Slide eight well slide (Ibidi) at 12°C and lasted for 24 hr during which live microscopy was performed. Latrunculin A (Sigma-Aldrich) was used at final concentration of 50 µM from a stock of 20 mM in DMSO. CK666 (Sigma-Aldrich) was used at a final concentration of 100 µM from a stock of 10 mM in DMSO. SMIFH2 (Sigma-Aldrich) was used at a final concentration of 40 µM from a stock of 10 mM in DMSO. Blebbistatin (Sigma-Aldrich) was used at a final concentration of 1.25 µM from a stock of 2.5 mM in DMSO. MBC (Sigma-Aldrich) was used at final concentration of 130 µg/ml from a stock of 2.5 mg/ml in dimethyl sulfoxide (DMSO).

## Image analysis

Image analysis was done using ImageJ software (version 1.52) (*Schneider et al., 2012*). For measurements of cell diameter in *Figure 1E* and *Figure 1—figure supplement 1A*, we cropped movies to ensure having a single cell per movie. We then transformed the movies into binaries to ensure later segmentation. We then used particle analysis function in ImageJ with a circularity parameter set to 0.65–1 to quantify measure cell perimeter. As cells are spherical, we computed cell diameter as:

For quantification of fraction of cells in each stage of cellularization *Figure 2—figure supplement 1B*, we used the ObjectJ plugin in ImageJ (National Institutes of Health).

All Figures were assembled with Illustrator CC 2017 (Adobe).

# Acknowledgements

We thank Sophie Martin, Aaron New, Daniel Richter and Sébastien Wielgoss for discussion and comments on the manuscript, Takaaki Kai for discussion on the ichthyosporean development, Eduard Ocaña for advice on the phylostratigraphic analysis, and Meritxell Antó for technical support. We also acknowledge the UPF Flow Cytometry Core Facility for assistance with flow cytometry, CRG Advanced light microscopy unit for support with confocal imaging, and the CRG Genomics Unit for mRNA library preparation and Illumina sequencing. This work is dedicated to the memory of Arthur Haraldsen, our dear friend and colleague, who tragically passed away during the final stages of the manuscript preparation.

This work was funded by European Research Council Consolidator Grant (ERC-2012-Co −616960) to IR-T; MEXT KAKENHI (grant Nos. 221S0002 to AT; grants no 26891021 and 16K07468 to HS); and a Young Research Talents grant from the Research Council of Norway (grant no. 240284) to JB. OD was supported by a Swiss National Science Foundation Early PostDoc Mobility fellowship (P2LAP3_171815) and a Marie Sklodowska-Curie individual fellowship (MSCA-IF 746044). AO was supported by a Marie Sklodowska-Curie individual fellowship (MSCA-IF 747086).

# Additional information

## Funding

| Funder | Grant reference number | Author |
|---|---|---|
| European Research Council | Consolidator Grant ERC-2012-Co-616960 | Iñaki Ruiz-Trillo |
| Ministry of Education, Culture, Sports, Science and Technology | MEXT KAKENHI 221S0002 | Atsushi Toyoda |
| Ministry of Education, Culture, Sports, Science and Technology | MEXT KAKENHI 26891021 | Hiroshi Suga |

| Research Council of Norway | Young Research Talents grant 240284 | Jon Bråte |
| Swiss National Science Foundation | P2LAP3_171815 | Omaya Dudin |
| H2020 Marie Skłodowska-Curie Actions | Individual fellowship MSCA-IF 746044 | Omaya Dudin |
| H2020 Marie Skłodowska-Curie Actions | Individual fellowship MSCA-IF 747086 | Andrej Ondracka |

The funders had no role in study design, data collection and interpretation, or the decision to submit the work for publication.

## Author contributions

Omaya Dudin, Conceptualization, Data curation, Formal analysis, Funding acquisition, Validation, Investigation, Visualization, Methodology, Writing—original draft; Andrej Ondracka, Conceptualization, Resources, Data curation, Software, Formal analysis, Funding acquisition, Validation, Investigation, Visualization, Methodology, Writing—original draft; Xavier Grau-Bové, Data curation, Software, Formal analysis, Investigation, Visualization, Methodology, Writing—review and editing; Arthur AB Haraldsen, Data curation, Software, Formal analysis, Investigation, Methodology, Writing—review and editing; Atsushi Toyoda, Resources; Hiroshi Suga, Resources, Formal analysis, Methodology, Writing—review and editing; Jon Bråte, Resources, Formal analysis, Supervision, Investigation, Visualization, Methodology, Writing—review and editing; Iñaki Ruiz-Trillo, Supervision, Funding acquisition, Writing—review and editing

## Author ORCIDs

Omaya Dudin (iD) https://orcid.org/0000-0002-6673-3149
Andrej Ondracka (iD) https://orcid.org/0000-0003-4193-6027
Xavier Grau-Bové (iD) http://orcid.org/0000-0003-1978-5824
Atsushi Toyoda (iD) http://orcid.org/0000-0002-0728-7548
Jon Bråte (iD) http://orcid.org/0000-0003-0490-1175
Iñaki Ruiz-Trillo (iD) https://orcid.org/0000-0001-6547-5304

## Decision letter and Author response

Decision letter https://doi.org/10.7554/eLife.49801.sa1
Author response https://doi.org/10.7554/eLife.49801.sa2

# Additional files

## Supplementary files

• Transparent reporting form

## Data availability

Sequencing data have been deposited at the following locations: S. arctica genome assembly - BioProject number PRJDB8476; S. arctica transcriptomes - PRJEB32922 (ERP115662) on European Nucleotide Archive.

The following datasets were generated:

| Author(s) | Year | Dataset title | Dataset URL | Database and Identifier |
| --- | --- | --- | --- | --- |
| Suga H, Toyoda A, Ruiz-Trillo I | 2018 | Sphaeroforma arctica genome assembly | https://www.ncbi.nlm.nih.gov/bioproject/?term=PRJDB8476 | NCBI BioProject, PRJDB8476 |
| Ondracka A, Dudin O, Ruiz-Trillo I | 2018 | Dynamics of transcription during the coenocytic cycle of Sphaeroforma arctica | https://www.ebi.ac.uk/ena/data/view/PRJEB32922 | European Nucleotide Archive, PRJEB32922 |

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
