## [Decision Letter]

**Acceptance summary:**

Multicellular life has arisen many independent times across the tree of life, including in plants and animals. Apart from being fascinating from an evolutionary point of view, the phenomenon of multicellularity raises fundamental cell-biological questions. Some of the complex developmental processes we typically think of as unique to plants and animals are also seen in their unicellular relatives. Is this due to convergence, or is it because such processes are truly ancient, arising first in unicellular ancestors and eventually being re-purposed for a multicellular mode of living? If we could demonstrate that the same set of molecular players were involved in both unicellular and multicellular linages, this would essentially rule out convergence and support the latter scenario. In this paper the authors study a widespread mode of animal development called coenocytic growth, in which a single cell goes through repeated rounds of nuclear replication before eventually splitting into a large number of new cells. Coenocytic growth also occurs the ichthyosporean *S. arctica*, a unicellular relative of animals. The authors demonstrate, using cell-biological investigations, transcriptional profiling, and pharmacological inhibition, that cellularization in *S. arctica* is driven by a network of actin, myosin, and their regulatory partners, and cell-cell adhesion proteins, just as is observed in animals. This points to an early evolutionary origin of cellularization, prior to true animal multicellularity.

**Decision letter after peer review:**

Thank you for submitting your article "A unicellular relative of animals generates a layer of polarized cells by actomyosin-dependent cellularization" for consideration by *eLife*. Your article has been reviewed by three peer reviewers, including Mukund Thattai as the Reviewing Editor and Reviewer #1, and the evaluation has been overseen by K VijayRaghavan as the Senior Editor.

The reviewers have discussed the reviews with one another and the Reviewing Editor has drafted this decision to help you prepare a revised submission.

Summary:

Cellularization of an epithelial coenocyte is a widespread mode of animal development. Coenocytic growth also occurs in plants and unicellular organisms. The mechanisms of cellularization are distinct across lineages, suggesting a pattern of convergent evolution. In this paper the authors study cellularization of a coenocyte in the ichthyosporean *S. arctica*, a unicellular relative of animals. They demonstrate, using cell-biological investigations, transcriptional profiling, and pharmacological inhibition, that cellularization is driven by a contractile network of actin, myosin, and their regulatory partners, and cell-cell adhesion proteins, just as is observed in animals. Together, the authors provide a convincing case for the presence of a self-organized, clonally-generated, polarized layer of cells in a unicellular relative of animals. Their results argue against convergence and point to an early evolutionary origin of coenocytic cellularization, prior to true animal multicellularity.

Essential revisions:

The reviewers find this study to be striking, exciting and well conducted. However, they request that the following points be addressed in a revised submission prior to acceptance.

1) Datasets: The article reports conclusions drawn from new genome sequences of *S. arctica* and *S. tapetis*. The sequencing, genome assembly, gene annotation, and RNA-seq data are not provided along with the manuscript. Links to primary data should be provided in the revision, to the accepted standard that ought to accompany the reporting of a newly-sequenced genome. (There is also no GitHub link provided for analysis code.)

2) Literature: Throughout the paper the authors refer to the work in *Drosophila* as this is indeed the model where cellularization has been chiefly studied. It is noteworthy that the authors consistently refer to a single review by Mazumdar and Mazumdar (who interestingly never worked on cellularization) but never cite the primary literature (mostly from the Wieschaus lab), with the exception of a single paper (Hunter and Wieschaus). This presents a rather skewed reference to the literature, and omits the contribution of membrane dynamics (endo/exocytosis) and its interaction with the actin cytoskeleton (i.e. membrane tension) in cellularization, which has also been a subject of intense investigation in standard cytokinesis. More generally the idea of membrane reservoirs at the cell surface (and in organelles/vesicles) has been thoroughly addressed (e.g. dating back from Erickson and Trinkaus, 1976) but is absent from analysis and discussions. The cited literature should be broadened to address these issues.

3) Statistical support: The authors state that "overall age of the transcriptome across the life cycle revealed an hourglass pattern". Although visually this seems to be the case, it would strengthen the point if the authors could provide statistical evidence by performing an hourglass test for the transcriptome age index. (For example, tools for such a test are provided in Drost et al., 2015.)

4) Materials and methods: The authors should provide, in Materials and methods, sufficient technical details underlying phylostratigraphic analysis so readers can reproduce the main results. How were splice variants handled when passing proteome sequences to orthofinder? Which output of orthofinder was used? The core orthologs (unicopy 1-1 orthologs) or the pairwise orthologs between all pairwise species comparisons? How many orthogroups were found? The authors could provide either reproducible analysis scripts, or the orthofinder output and phylostratum categorization of genes used to be able to reproducibly compute TAI values (as supplementary data).

---

## [Author Response]

Essential revisions:The reviewers find this study to be striking, exciting and well conducted. However, they request that the following points be addressed in a revised submission prior to acceptance.1) Datasets: The article reports conclusions drawn from new genome sequences of S. arctica and S. tapetis. The sequencing, genome assembly, gene annotation, and RNA-seq data are not provided along with the manuscript. Links to primary data should be provided in the revision, to the accepted standard that ought to accompany the reporting of a newly-sequenced genome. (There is also no GitHub link provided for analysis code.)

We have provided links to all of the requested genome files, as well as the transcriptome analysis code in the Materials and methods section.

2) Literature: Throughout the paper the authors refer to the work in Drosophila as this is indeed the model where cellularization has been chiefly studied. It is noteworthy that the authors consistently refer to a single review by Mazumdar and Mazumdar (who interestingly never worked on cellularization) but never cite the primary literature (mostly from the Wieschaus lab), with the exception of a single paper (Hunter and Wieschaus). This presents a rather skewed reference to the literature, and omits the contribution of membrane dynamics (endo/exocytosis) and its interaction with the actin cytoskeleton (i.e. membrane tension) in cellularization, which has also been a subject of intense investigation in standard cytokinesis. More generally the idea of membrane reservoirs at the cell surface (and in organelles/vesicles) has been thoroughly addressed (e.g. dating back from Erickson and Trinkaus, 1976) but is absent from analysis and discussions. The cited literature should be broadened to address these issues.

We agree. We have expanded the literature and references to cover zygotically transcribed genes and membrane remodeling. We now also emphasize that homologs of zygotically transcribed genes (mostly done by Wieschaus lab) regulating cellularization in *Drosophila* (such as *nullo* and *slam*) are absent in *S. arctica.* Finally, we have added Figure 5—figure supplement 1 showing the expression of homologs of Rab5 and Rab11 which regulate membrane trafficking in *Drosophila* and are important for cellularization. Despite being highly expressed (for most), their expression profiles appears to be constant. This might suggest that membrane trafficking does not only intervene during cellularization but also is important during coenocytic development. However, in absence of further investigation, that we believe is beyond the scope of this study, we cannot speculate on the role of membrane trafficking in the cellularization of *S. arctica*.

3) Statistical support: The authors state that "overall age of the transcriptome across the life cycle revealed an hourglass pattern". Although visually this seems to be the case, it would strengthen the point if the authors could provide statistical evidence by performing an hourglass test for the transcriptome age index. (For example, tools for such a test are provided in Drost et al., 2015.)

We thank the reviewers for pointing out these statistical tests. We have implemented the tests, and we indeed find statistical support for the hourglass pattern for both replicates.

4) Materials and methods: The authors should provide, in Materials and methods, sufficient technical details underlying phylostratigraphic analysis so readers can reproduce the main results. How were splice variants handled when passing proteome sequences to orthofinder? Which output of orthofinder was used? The core orthologs (unicopy 1-1 orthologs) or the pairwise orthologs between all pairwise species comparisons? How many orthogroups were found? The authors could provide either reproducible analysis scripts, or the orthofinder output and phylostratum categorization of genes used to be able to reproducibly compute TAI values (as supplementary data).

We have expanded the Materials and methods section on the orthofinder and phylostratigraphic analysis to address the reviewer’s concerns. Furthermore, in addition to the file of all orthogroups, which was already uploaded as source data, we now include a categorization of genes by gene age according to Dollo parsimony, as a source data file (Figure 4—source data 5).